# Non-Coding RNAs: Novel Regulators of Macrophage Homeostasis in Ocular Vascular Diseases

**DOI:** 10.3390/biom14030328

**Published:** 2024-03-10

**Authors:** Qiuyang Zhang, Qing Liu, Siguo Feng, Xiumiao Li, Qin Jiang

**Affiliations:** 1Affiliated Eye Hospital, Nanjing Medical University, Nanjing 210029, China; zhangqy@njmu.edu.cn (Q.Z.); liuqing67@stu.njmu.edu.cn (Q.L.); fsg@stu.njmu.edu.cn (S.F.); 2The Fourth School of Clinical Medicine, Nanjing Medical University, Nanjing 210029, China

**Keywords:** non-coding RNA, macrophage, pathological angiogenesis

## Abstract

Ocular neovascularization can impair vision and threaten patients’ quality of life. However, the underlying mechanism is far from transparent. In all mammals, macrophages are a population of cells playing pivotal roles in the innate immune system and the first line of defense against pathogens. Therefore, it has been speculated that the disfunction of macrophage homeostasis is involved in the development of ocular vascular diseases. Moreover, various studies have found that non-coding RNAs (ncRNAs) regulate macrophage homeostasis. This study reviewed past studies of the regulatory roles of ncRNAs in macrophage homeostasis in ocular vascular diseases.

## 1. Introduction

Angiogenesis, under pathological conditions, can lead to vascular disfunction, especially in the eyes. Ocular neovascularization is a pathological hallmark in a variety of ocular diseases, such as age-related macular degeneration (AMD), diabetic retinopathy (DR) and retinopathy of prematurity (ROP), all manifesting impaired visual acuity that threatens the patient’s life quality [1]. Blocking pathological neovascularization is key to the treatment of these diseases. However, current treatment modalities are far from satisfactory, which necessitates the discovery of new mechanisms of ocular vascular diseases.

Macrophages take on a high profile in the innate immune system. Perceiving specific stimuli, the homeostasis among macrophages is broken, and they may differentiate into different subgroups to undertake specific immune tasks in various phenotypes. This process, also called macrophage polarization, is essential for tissue repair and system homeostasis [2,3]. Macrophages are mainly divided into two polarized types, namely classically activated (M1) macrophages and alternatively activated (M2) macrophages. M1 macrophages exert a “pro-inflammatory” effect to damage tissues and a “killing” effect to fight against the invasion of pathogens. M2 macrophages assume “anti-inflammatory and repairing” roles in tissue repair, lipid metabolism, and tumor development [4,5,6]. A number of studies have confirmed that ocular vascular diseases may develop as macrophages polarize abnormally, during which process a variety of active cytokines are secreted. This pathogenic mechanism may be exploited to design new diagnostic and therapeutic strategies.

In recent years, non-coding RNAs (ncRNAs) have become the focus of medical research. They can form complex interactions with other types of DNAs/RNAs, proteins, lipids, etc. Their regulatory roles in multiple cellular processes, including immune responses, have been clarified. Moreover, various studies have found that ncRNAs also serve to maintain macrophage homeostasis [7]. Here, we reviewed the regulatory roles of ncRNAs in macrophage homeostasis in ocular vascular diseases.

## 2. Macrophage Homeostasis and Ocular Vascular Diseases

Macrophages originate from bone marrow monocytic stem cells. In the bone marrow, original monocytes proliferate and differentiate into monocytes and then migrate along the blood stream to different tissues until developing into mature macrophages. In the innate immune system, macrophages constitute the first line of defense against external infections. Firstly, they serve as scavengers to engulf the cell debris generated after trauma or infection [8]. In tissue remodeling and innate immunity, the surface receptors of macrophages can transform into antigen-presenting cells (APCs) to activate the adaptive immune system [9]. In addition, macrophages act as myeloid-derived suppressor cells (MDSCs) to inhibit immune response or induce immune tolerance [10,11]. However, studies have shown that macrophages may regulate angiogenesis as a tumor progresses [12] and that macrophages can mediate the process of neovascularization [13,14], suggesting the necessity of further investigations into the critical role of macrophages in neovascularization (Figure 1).

Macrophages of the M1 type, also known as proinflammatory type, can damage tissues and inhibit tumor progression; the M2 type, also known as the anti-inflammatory type, can repair tissues and promote tumor occurrence, proliferation, and metastasis. IL-1β, IL-12, IL-23, IFN-γ, LPS, and TNF-α can induce polarization into M1 macrophages marked by CCL3, CCL5, CD80, CCR7, and iNOS. Other molecules, such as IL-4, IL-10, IL-13, and TGF-β, induce polarization in M2 macrophages that can be marked by CCL22, CD206, and CD163. With changes in the tumor microenvironment (TME), these two types of macrophages undergo mutual differentiation. As the balance between M1 and M2 macrophage differentiation is disrupted, inflammatory diseases and eye diseases may arise [15,16,17] (Figure 2). Therefore, it is of great significance to explore the key molecules and their regulatory mechanisms in macrophage polarization so that better tools can be created for the diagnosis and treatment of ocular vascular diseases.

## 3. Functions of NcRNAs

Ever-upgrading high-throughput technologies have made transcriptomic analysis more fruitful, leading to the discovery of a class of ncRNAs, which account for 98% of human gene transcripts [18]. Most ncRNAs are divided into two groups based on their sizes: small (<200 bp) and long non-coding RNAs (>200 bp) [19]. Small ncRNAs (sncRNAs) are further divided into microRNAs (miRNAs), small nucleolar RNAs (snoRNAs), small interfering RNAs (siRNAs), small nuclear RNAs (snRNAs), transfer RNAs (tRNAs), and piwi-interacting RNAs (piRNAs) [20]. Non-coding RNAs can form complex interactions with other types of DNAs/RNAs, proteins, lipids, etc. Their regulatory roles in multiple cellular processes, including immune processes, have been intensely studied [21,22,23,24]. At present, the below-mentioned types of ncRNAs are considered to be research hotspots (Figure 3).

### 3.1. MiRNAs

MiRNAs are a class of highly conserved short ncRNAs containing about 19-25 nucleotides. The miRNA microarray is performed to screen differentially expressed miRNAs. To detect the expression level of miRNA, RNA is extracted using miRNA isolation kit and detected by RT-qPCR [25]. miRNAs silence target gene expression by recruiting ribose nucleoprotein (RNP) complexes to bind to complementary RNA chains, cutting and degrading the mRNAs or other non-coding RNAs. The production of miRNAs in cells follows strict temporal and spatial rules. After primary transcription, pri-miRNA is produced; then, the precursor miRNA (pre-miRNA) is cleaved by the RNaseIII–drosha enzyme, and the mature miRNA is cleaved by another RNaseIII–Dicer enzyme. miRNAs regulate gene expression by incorporating the RNA-induced silencing complex (RISC) into the 3′ untranslated regions (3′-UTR) of the gene [26]. miRNAs modulate gene expression by targeting splicing factors (SFs) or RNA-binding proteins (RBPs) [27,28] or specifically targeting alternatively spliced transcripts [29]. One miRNA can target multiple mRNAs and affect the expression of genes related to functional interacting pathways.

Abnormal expression of miRNAs is associated with the occurrence and development of various diseases. Mechanistic research has proven that miRNAs are involved in many biological processes, such as cell proliferation, apoptosis, inflammatory responses, etc. [30]. Macrophages are functionally dynamic cells derived from monocytes. Functional macrophage polarization may initiate as a transcriptional response is induced by signals from the microenvironment, including signal transducer and the activator of transcription 1 (STAT1), interferon-regulatory factor 5 (IRF5), STAT6, and peroxisome proliferator-activated receptor-γ (PPARγ) [31,32]. These transcription factors can be regulated directly by miRNAs.

The role of miRNAs in regulating innate immune response and macrophage function has been widely recognized. For instance, miR-345-3p promotes the polarization of M1 macrophages to the M2 phenotype by inhibiting the MAP3K1 and NF-κB pathways [33]. In addition, miR-146a regulates macrophage polarization by inhibiting the Notch1 signaling pathway [34]. Moreover, miR-217 can target IL-6 and regulate the JAK3/STAT3 signaling pathway, eventually inhibiting M2 polarization [35]. Based on these miRNAs, new intervention strategies have been proposed to regulate the proliferation, differentiation, and activity of macrophages in biological responses [36].

### 3.2. LncRNAs

LncRNAs, a class of transcripts containing more than 200 nucleotides, function actively in the immune system, epigenetics, disease development, and cell differentiation. According to their positions in the genome, lncRNAs are classified into sense lncRNAs, antisense lncRNAs, intron lncRNAs, bidirectional lncRNAs, and intergenic lncRNAs [37]. According to their modes of interaction with other elements, lncRNAs are divided into signal lncRNAs, decoy lncRNAs, guide lncRNAs, and scaffold lncRNAs [38]. To identify lncRNA, RNA sequencing technology and RT-qPCR can be used [39]. RNA in situ hybridization can be used to detect the distribution of lncRNA in cells or tissues [40]. LncRNAs are located in both the cytoplasm and nucleus, suggestive of their broad functions. LncRNAs can act as miRNA sponges to regulate mRNA expression mediated by miRNAs [41]. LncRNAs also appear as miRNA precursors [42] or scaffolds for ribonucleoproteins [43]. In addition, lncRNAs serve as a decoy [44] or guide for RNA to control gene expression [45]. Moreover, lncRNAs are also involved in gene regulation by facilitating chromatin loop formation [46]. Otherwise, their expression levels vary by cell type, indicating their potential roles in cellular biological process [47]. Accumulated evidence has demonstrated their diverse roles in multiple functional processes, including protein synthesis and RNA maturation. In these processes, lncRNAs act as signals, decoys, or transporters and silence gene expression through reshape chromatins. In recent years, increasing studies have found that lncRNAs are also involved in macrophage differentiation or polarization [4,48]. For example, lncRNA-*Mirt2* promotes M2 polarization by inhibiting the Lys63 (K63) ubiquitination of TRAF6 [49]. LncRNA-*NEAT1* can bind to miR-125a-5p and upregulate expression of TRAF6 and TAK1, causing M1 polarization [50]. Moreover, lncRNA- *IGHCγ1* promotes expression of TLR4 by acting as a ceRNA to bind with miR-6891-3p, which aggravates the TLR4-induced inflammatory response of macrophage [51].

### 3.3. CircRNAs

CircRNAs, another class of non-coding RNAs, are named for their closed covalent ring structure and widely found in eukaryotes. circRNAs are derived from the exon or intron region of genes and highly conserved across species due to their structural stability. CircRNA microarray analysis is used commonly for circRNA detection. For circRNA analysis, RT-qPCR is used for determining the expression level of circRNA, and primers for circRNAs were designed to cross the back-splice junction site [52]. Otherwise, the RNase R assay is usually conducted to detect the stability of circRNA, which is used to verify the characteristics of the circular structure [53].

CircRNAs are generated in the nucleus, but most of them serve functions in the cytoplasm [54]. Abnormally expressed circRNAs induce multiple cellular functions, including proliferation, migration, resistance to apoptosis, and angiogenesis [55,56]. In recent years, circRNAs have become a hot topic in RNA research due to their high specificity and complex regulatory network. circRNAs target genes to regulate transcription, splicing, and chromatin interactions [57]. The evidence has shown that circRNAs partake in the development, induction, and activity of macrophages. For example, Zhang and colleagues found that a total of 189 circRNAs were differentially expressed between M1 and M2 macrophages [52]. Moreover, circ*ITGB6* increases the stability of FGF9 mRNA by binding to IGF2BP2, which subsequently promotes M2 polarization and, consequently, induces drug resistance in ovarian cancer cells [58]. Furthermore, circ*KIAA0391* has been reported to act as a miRNA sponge for the miR-512-5p/MITA axis, promoting the proinflammatory polarization and apoptosis of macrophages [59]. Due to their inherent stability, circRNAs have exhibited potential as biomarkers or therapeutic targets for immune disorders [60].

## 4. NcRNAs Regulate Macrophage Homeostasis in Ocular Pathological Angiogenesis-Related Disorders

### 4.1. AMD

AMD, a common form of macular degeneration, is manifested through the growth of drusen, which can decline central vision [61]. Drusen develops with the accumulation of extracellular lipids, proteins, and inflammatory factors, mainly being distributed between the RPE and Bruch’s basement membrane [62,63]. AMD can be divided into two types. Wet AMD is characterized by immature choroidal neovascularization (CNV) that appears below the RPE layer or breaks through the RPE layer into the outer retina. Atrophic or dry AMD, the other type, is characterized by progressive map atrophy and progressive enlargement of the RPE and photoreceptor cell degeneration area [61].

Under normal physiological conditions, mononuclear cells are absent in the photoreceptor layers of the retina and the subretinal space, maintaining an inherent immunosuppressive state [64]. However, in AMD, monocyte infiltration and microglial activation occur in both layers at any stage of the disease [64,65], suggesting that the infiltration and activation of retinal macrophages and their secreted proinflammatory factors may lead to to the loss of photoreceptor cells and the aggravation of the progression of AMD.

Macrophages have been proven to be the main immune cells responsible for the progression of CNV in wet AMD [66]. Macrophage chemoattractant protein-1 (MCP-1), also known as CCL2, is a key chemokine for monocyte recruitment. As a chemokine of M1 macrophages, it is also involved in inflammation induction [67]. Serum MCP-1 levels in patients with wet AMD are significantly higher than those in age- and sex-matched controls [68]. After analyzing and comparing peripheral blood samples collected from patients with wet AMD, dry AMD, and age-matched controls, it has been found that the levels of a variety of cytokines, such as CD163 (M2 macrophage-specific marker), show significant between-group differences and are related to active vascular leakage in patients with wet AMD, suggesting that these cytokines may be used as diagnostic biomarkers of AMD [66]. IL-10 and senescence are two major regulators involved in the polarization, activation, and abnormal function of macrophages in a laser-induced mouse CNV model. Specific IL-10 knockout in eyes induces the differentiation of macrophages into an anti-angiogenic subtype, manifested by elevated levels of TNF-α, IL-6, and IL-12 (type M1).

Studies have uncovered that the above-mentioned macrophages hijack the apoptosis-related factor ligand FasL to promote endothelial cell apoptosis and inhibit the progression of CNV. In M1-type macrophages and necrotic retinal tissues, the express of FasL is elevated. On the other hand, IL-10 can polarize macrophages into a pro-angiogenic phenotype, thus reducing the levels of TNF-α, IL-6, IL-12, and FasL (M2 type). In addition, researchers have also found that IL-10 levels rise gradually with age, suggesting that age is not only a risk factor for AMD but may also be a contributor to the polarization of M2-type cells. Interestingly, it was found that the function of macrophages in CNV is regulated by the mTORC1 signaling pathway. The inhibition of mTORC1 activity can reduce the M1-oriented polarization and enhance the M2-oriented polarization of macrophages, while activation on mTORC1 has the opposite results. Moreover, mTORC1 in macrophages can be pinpointed to manipulate the expression of cytokines related to CNV progression in RPE cells, including PEDF, MMP9, IL-1β, and MCP-1 [69]. Zhu et al. found that the expression of IL-17A increases in retinal neovascularization, and the knockout of IL-17A turns macrophages toward M2 and decreases the expression of M1-related cytokines. The intraocular injection of IL-17 neutralizing antibodies inhibits choroidal and retinal neovascularization. Macrophage supernatant treated with IL-17 can promote the proliferation and tubulation of vascular endothelial cells and the expression of VEGFR-1 and VEGFR-2 [70]. Liu et al. found that HPCL-03, an antitumor drug, can downregulate the VEGF/VEGFR2/phosphoinosidine-3-kinase (PI3K)/protein kinase (AKT)/NFκB signaling pathway in endothelial cells, reduce the secretion of CCL2, and inhibit VEGF/VEGFR2-induced M1 polarization. It is suggested that this drug can inhibit CNV by interrupting the interactions between macrophages and endothelial cells [71].

Recent studies have tried to identify ncRNAs that regulate macrophage homeostasis in AMD. One study showed the roles of macrophage senescence and cholesterol homeostasis in AMD, advocating that an miR-714-*FDFT1* network may steer the development of AMD by regulating cholesterol homeostasis in aging macrophages [72]. Another study explored the effects of miRNAs on the neovascularization and inflammatory response in AMD. The researchers found that miR-142-3p is upregulated in a laser-induced AMD model, and appropriate interference with its expression lowers the degree of pathological neovascularization and inflammatory response. In addition, miR-142-3p can activate local microglial cells, which are also called macrophages in the retina, and induce their differentiation into a proinflammatory state [73]. Mononuclear phagocytes, including monocytes/macrophages, cooperate intricately in the pathogenesis of AMD. To elucidate the role of monocytes in AMD, researchers have conducted transcriptomic analysis of monocytes in the peripheral blood of patients with AMD. They profile gene expression in peripheral blood monocytes and the levels of monocyte chemotactic factor receptors. In this study, multiple miRNAs are differentially expressed, among which miR-283 is downregulated in nvAMD patients; VEGF-A is predicted to be its target, and reducing VEGF-A expression by inhibiting miR-283 may have a protective effect on neovascularization [74]. Lin et al. found that miR-150 can change the transcriptomic profiles of macrophages in elderly mice, leading to abnormal lipid transport and metabolism in AMD. Mechanically, miR-150 directly targets stearoyl-CoA desaturase-2, a factor that mediates macrophage-related inflammatory response and pathological angiogenesis independent of VEGF [25]. Another study shows that miR-505-5p can negatively regulate the expression of transmembrane protein 229B, thereby promoting M2 polarization, VEGF expression, and CNV formation in CNV mice [75]. In addition to the direct effect on macrophages, the researchers found that extracellular vesicles secreted by the retinal pigment epithelium (RPE) epithelium carry miRNAs that can interact with macrophages to increase the levels of MCP-1, IL-6, IL-8. and VEGF and inhibit the secretion of pigment epithelium-derived factor (PEDF) and TNF-α from macrophages and RPE cells. miR-494-3p may be a candidate molecular target for the diagnosis and treatment of AMD [76].

In addition to miRNAs, lncRNAs have an effect on macrophage homeostasis in AMD. Zhang et al. sequenced the expression of lncRNAs in the choroid–sclera complex of laser-induced AMD mice, finding them in the aqueous humors of nvAMD patients. The results show that lncRNA *H19* is significantly highly expressed in the aqueous humors of patients, and the inhibition of its expression impedes the M2 polarization of macrophages [77]. In addition, bioinformatics analyses showed that circRNA7329 may regulate the downstream *Stearoyl-CoA desaturase* (*SCD*) gene through miR-9 to promote the inflammatory responses of macrophages and pathological angiogenesis in AMD, but the specific mechanism needs to be further clarified [78].

### 4.2. DR

Vascular dysfunction may accompany diabetes mellitus (DM), which is characterized by hyperglycemia caused by insulin resistance. Homeostasis imbalance caused by pathological environment, such as hyperglycemia, ischemia, and hypoxia, can activate microglia/macrophages and may result in retinal inflammation and DR [79]. DR, one most serious microvascular complications of DM, accounts for progressive vascular disfunctions, including high vascular permeability, capillary microaneurysms, capillary degeneration, and neovascularization. These vascular disfunctions cause chronic inflammation, further aggravating the chronic ischemia and hypoxia of the retina.

Liou et al. have found that miR-146b-3p inhibits its own expression and activity by binding to the 3′-UTR break of adenosine deaminase 2 (ADA2) and that stronger ADA2 activity is associated with higher levels of macrophage activation and TNF-α release in diabetic patients with retinal inflammation. This suggests that miR-146b-3p may be a therapeutic target for the early diagnosis of and intervention in DR [80]. Based on the same group, another study has also reconfirmed that ADA2 is significantly activated in the vitreous body of DR patients, stimulation with glycated albumin can increase the expression levels of proinflammatory factors TNF-α and IL-6 in macrophages, and the supernatant from treated macrophages significantly increases the permeability of endothelium. These changes can be reversed by miR-146b-3p, showing that the inhibitory effect of miR-146b-3p on ADA2 may be replicated to clinically protect against the destruction of the blood–retinal barrier in DR [81]. Another study has found that miR-125a-5p inhibits the infiltration and activation of macrophages in the retina of DR patients by targeting *Ninj1*, a mechanism that may be maneuvered to counter retinal vascular disease [82].

LncRNAs can induce the polarization of macrophages to control their functions. By analyzing the transcriptomes of bone marrow macrophages in diabetic db/db mice, researchers find that macrophages grow into proinflammatory and pro-fibrotic phenotypes at the transcriptional level in a diabetic condition. Interestingly, lncRNA *E330013P06* is upregulated in macrophages of type 2 diabetic mice, which increases the expression of proinflammatory genes and enhances their responses to proinflammatory signals. lncRNA *E330013P06* helps macrophages to differentiate into proinflammatory phenotypes [83]. lncRNA *DRAIR* is downregulated in mononuclear cells of patients with type 2 diabetes, and overexpression of *DRAIR* increases the abundance of anti-inflammatory macrophages and decreases those of proinflammatory genes. Mechanically, *DRAIR* regulates the differentiation of macrophages into inflammatory phenotypes through epigenetic mechanisms, thereby affecting the progression of diabetes and its complications [84]. However, the relationship between lncRNAs and macrophage homeostasis in DR necessitates further in-depth studies.

### 4.3. ROP

ROP, a vascular proliferative retinal disease, mainly attacks premature infants, causing blindness and visual impairment [85]. The course of ROP is biphasic: retinal blood vessel occlusion induced by hyperxia in the initial phase, and blood vessel dysplasia induced by ischemia and hypoxia in the second phase, followed by neovascularization and neurodegeneration [86]. ROP involves etiological factors in both prenatal and postnatal stages. Currently, inflammation is believed as a leading factor in ROP pathogenesis, because it interferes with retinal vascularization, and oxidative stress ensuing inflammation also plays a pathological role [87,88]. Previous studies have concluded that microglia/macrophages are activated throughout the process of ROP in hyperoxic and hypoxic phases [89]. Moreover, it has been found that the progression of oxygen-induced retinopathy (OIR) in mice is closely related to the differentiation of microglia/macrophages [90].

MiR-155 knockout mice show mild vascular defects, while the overexpression of miR-155 boosts endothelial cell proliferation and increases the number and activity of microglia. miR-155 expression is upregulated in OIR model mice, and miR-155 silencing reduces the activity of microglia, thus curbing the growth of abnormal blood vessels and potentiating the proliferation the retinal blood vessels after ischemic injury [91]. miR-24-3p may be a key molecule in inhibiting the expression of inositol-requiring enzyme 1α (IRE1α) in photoreceptor cells, leading to photoreceptor apoptosis [92]. Another study shows that the expression level of miR-24-3p climbs up in hypoxia-induced microglial exosomes. A number of studies have found that lncRNAs and circRNAs are abnormally expressed in OIR mice [93,94,95], but whether lncRNAs have a regulatory effect on macrophage homeostasis in the retina of OIR mice and hypoxia-induced retinal pathological neovasculation still needs sound evidence.

### 4.4. Ocular Tumor

Angiogenesis and immune protection are critical at the process of tumorigenesis. Angiogenesis serves to accelerate tumor invasion and the prevention of immune defenses. Ocular tumors are rare and involve a distinct ocular structure depending on the type of tumor. For example, retinoblastoma arises in retina, conjunctival melanoma involves the conjunctival epithelium, and uveal melanoma affects the region of the uveal tract [96]. Furthermore, the eye may be the metastatic sites of other primary tumors such as breast cancers, lung cancers, and cutaneous melanoma [97,98]. Macrophages play roles in immune regulation during inflammation, disease pathogenesis, and cancer development. Increasing evidence has revealed that macrophage polarization and infiltration are involved in cancer progression [99]. Both the M1 macrophage and M2 macrophage has regulatory roles during tumor development, with the M1 type inducing inflammatory responses and apoptotic death in cancer cells and the M2 type inhibiting immune responses and promoting tumor growth and angiogenesis [100,101]. In a specific type of ocular tumor such as uveal melanoma, a high density of macrophage infiltration is related to a worse prognosis [102]. It has been reported that miR-30a-5p-loaded nanodrug inhibited ocular melanoma by modulating M1 macrophage polarization [103]. However, the roles of other type of ncRNAs in ocular tumors by regulating macrophages functions are still uncertain, which led us to focus on these fields in the future.

## 5. Conclusions and Future Perspectives

The eyeball is a highly specialized, complex sensory organ containing a variety of cell and tissue types that require synergistic, fine-tuned regulation in order to obtain clear and complete visual images. Macrophages are present in all layers of the eye and regulate eye microenvironment, inflammation, and vascular remodeling. As the central inflammatory infiltrating cell, macrophages regulate disease processes by secreting multiple inflammatory cytokines. Additionally, macrophage polarization has been proved to be closely related to ocular angiogenetic pathogenesis. Thus, macrophages are key mediators between the immune system and pathological neovascularization, indicating that macrophages may be a desired therapeutic target of ocular vascular diseases.

With the help of high-throughput sequencing methods, rapid screening of the potential effectors of angiogenesis-relative disorders has been achieved and enabled a more comprehensive and deep understanding of these disorders. The inflammatory response and macrophage phenotype shift depend on the response to microenvironmental stimuli. Epigenetic mechanisms enhance macrophage diversity and plasticity, and ncRNAs become involved. The evidence that we discussed in this review suggests that ncRNAs epigenetically regulate macrophage homeostasis during the development of ocular neovascular diseases. However, the underlying mechanism still needs to be unraveled in further studies. The search for molecules and signaling pathways that effectively interfere with macrophage homeostasis should be continued to provide breakthrough treatments. Furthermore, the biological features of ncRNAs, such as multiple targeted ability, modifiability, or low molecule weight, suggest the possibility of drug discovery or diagnosis biomarker development. The ncRNA-based gene regulation-targeting macrophage functions provide hopeful insights into the treatment of ocular vascular disorders.

## Figures and Tables

**Figure 1 biomolecules-14-00328-f001:**
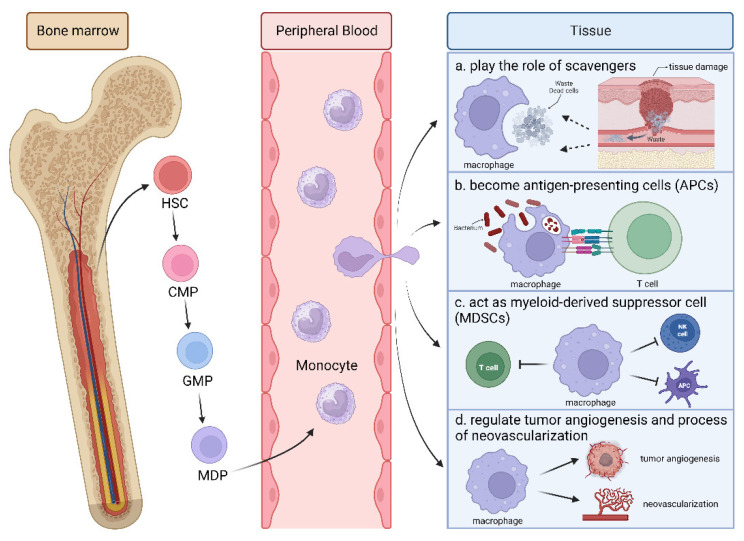
The origins and function of macrophages. Monocytes are continuously generated from hematopoietic stem cells (HSCs) via common myeloid progenitors (CMPs), macrophage progenitors (GMPs), and macrophage and dendritic cell (DC) precursors (MDPs) in the bone marrow. During inflammation, some of the monocytes are rapidly recruited to sites of injury, differentiating into macrophages to play multiple functions in the tissue. Firstly, serving as scavengers, macrophages can engulf waste such as pathogen and cell debris generated after trauma or infection. Secondly, with various surface receptors, macrophages can act as antigen-presenting cells (APCs) to activate the adaptive immune response. In tumors, macrophages can inhibit immune responses and induce immune tolerance as myeloid-derived suppressor cells (MDSCs). In addition, macrophages regulate tumor angiogenesis and neovascularization. This illustration was created with BioRender.com (https://biorender.com, accessed on 5 March 2024).

**Figure 2 biomolecules-14-00328-f002:**
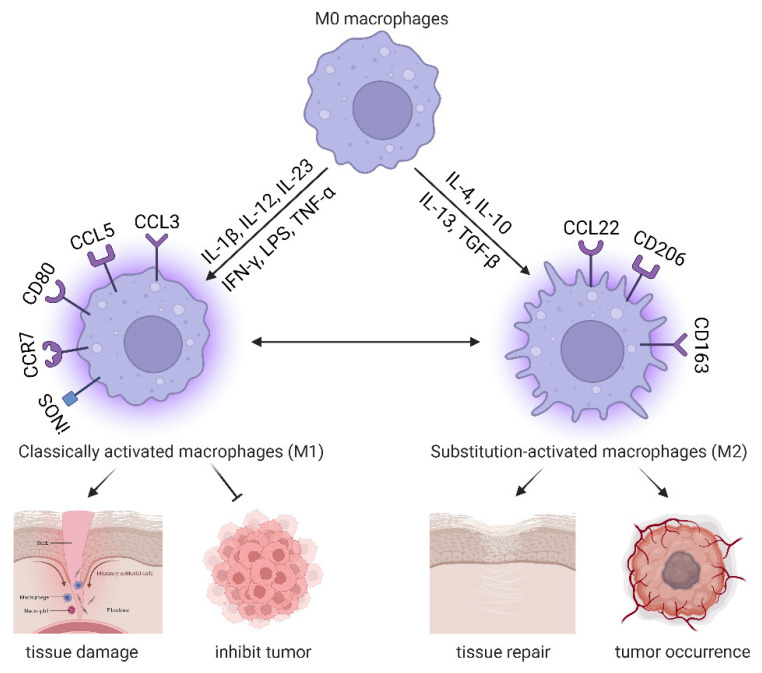
The main polarization states of activated macrophages. Classically activated proinflammatory M1 macrophages and alternatively activated anti-inflammatory M2 macrophages are the two canonical phenotypes of macrophages. On one hand, the polarization of M0 macrophages toward an M1 phenotype can be achieved via the induction of IL-1β, IL-12, IL-23, IFN-γ, LPS, and TNF-α involved in tissue damage and tumor inhibition. On the other hand, the polarization of M2 macrophages is activated by IL-4, IL-10, IL-13, and TGF-β involved in tissue repair and tumor occurrence. The M1 and M2 macrophages show distinctive cell surface and intracellular markers. CD80, CCL3, CCL5, CCR7, and iNOS are commonly recognized as M1 polarization markers, while CD206, CD163, and CCL22 are commonly recognized as M2 markers. This illustration was created with BioRender.com (https://biorender.com, accessed on 5 March 2024).

**Figure 3 biomolecules-14-00328-f003:**
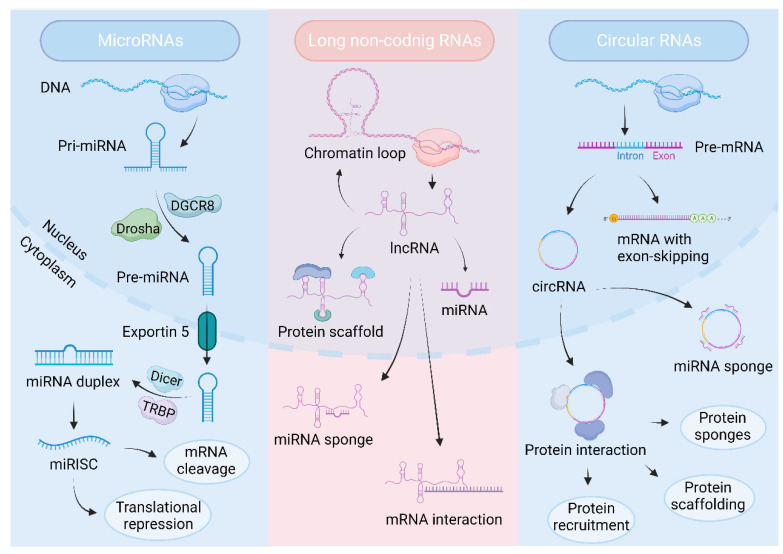
Biogenesis and function of miRNAs, lncRNAs, and circRNAs. miRNAs are transcribed from DNA into primary miRNAs (pri-miRNAs) that are processed into precursor miRNAs (pre-miRNAs) via DiGeorge syndrome critical region 8 (DGCR8) and Drosha. Pre-miRNAs leave the nucleus via exportin 5. In the cytoplasm the pre-miRNA is processed by Dicer and transactivation response element RNA-binding protein (TRBP), and a miRNA duplex is produced. miRNAs regulate mRNA cleavage and translational repression by incorporating into the RNA-induced silencing complex (RISC). LncRNAs regulate gene expression by facilitating chromatin loop formation. In addition, LncRNAs act as miRNAs precursors and scaffolds for ribonucleoproteins. Otherwise, lncRNAs regulate RNA function by acting as miRNA sponges and interacting with mRNAs. circRNAs are transcribed from the coding regions of the genome, including exons or introns. circRNAs mainly function in the cytoplasm as miRNA sponges or interact with proteins containing protein scaffolding, sponges, and recruitment. This illustration was created with BioRender.com (https://biorender.com, accessed on 5 March 2024).

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
