# Peer review of "Non-Coding RNAs: Novel Regulators of Macrophage Homeostasis in Ocular Vascular Diseases"

_biomolecules, 2024, doi:10.3390/biom14030328_

Round 1

Reviewer 1 Report

Comments and Suggestions for Authors

The authors provide an interesting review of novel regulators of macrophage homeostasis in ocular vascular diseases. Additional specific comments are included below.

# The title is clear and concise. It effectively conveys the focus of the review on the regulatory roles of non-coding RNAs in macrophage homeostasis in ocular vascular diseases.

# The introduction provides a comprehensive overview of the background and significance of the study. It effectively introduces the relevance of angiogenesis in ocular diseases and the role of macrophages. It could be beneficial for the readers to explicitly state the research question or hypothesis guiding the review.

# Figures 1 and 2, both figures are well-designed and contribute to the understanding of macrophage functions.

# Functions of ncRNA section: This section is informative and provides a good foundation for understanding the role of non-coding RNAs. However, it might be helpful to briefly introduce the main types of ncRNAs before delving into their functions.

# Regarding Section 4: This section is the core of the paper, presenting findings related to specific ocular diseases ROP, DR, and AMD. The information is detailed and well-organized. However, it would be valuable to explicitly state the methodology used for ncRNA identification and validation.

# The conclusion is clear and summarizes the main findings. It could be enhanced by emphasizing the potential clinical implications of the focus of the review and beneficial in suggesting directions for future research.

# The review could be more comprehensive by including discussions on the roles of these additional types of ncRNAs in macrophage homeostasis and ocular vascular diseases.

# Overall, the paper is well-written except few grammatical errors, and the information is presented logically. Additionally, addressing the minor points mentioned above would improve the overall clarity and impact of the paper.

Comments on the Quality of English Language

Few grammatical errors in the introduction

Author Response

Reviewer 1:

# The title is clear and concise. It effectively conveys the focus of the review on the regulatory roles of non-coding RNAs in macrophage homeostasis in ocular vascular diseases.

Response:

We are particularly grateful to the reviewers for their positive acknowledgment of our manuscript's title.

# The introduction provides a comprehensive overview of the background and significance of the study. It effectively introduces the relevance of angiogenesis in ocular diseases and the role of macrophages. It could be beneficial for the readers to explicitly state the research question or hypothesis guiding the review.

Response:

Thank you for the commendation of our introduction section.

# Figures 1 and 2, both figures are well-designed and contribute to the understanding of macrophage functions.

Response:

Thank you for the compliments for our figures.

# Functions of ncRNA section: This section is informative and provides a good foundation for understanding the role of non-coding RNAs. However, it might be helpful to briefly introduce the main types of ncRNAs before delving into their functions.

Response:

We deeply appreciate the reviewer's constructive feedback on the "Functions of ncRNA" section of our manuscript. Your suggestion to introduce the main types of non-coding RNAs (ncRNAs) before delving into their functions is indeed invaluable for ensuring clarity and comprehensiveness in our presentation. We have attempted to address this aspect in line 97-101.

# Regarding Section 4: This section is the core of the paper, presenting findings related to specific ocular diseases ROP, DR, and AMD. The information is detailed and well-organized. However, it would be valuable to explicitly state the methodology used for ncRNA identification and validation.

Response:

Thank you for your thorough review and valuable feedback on Section 4 of our manuscript. Regarding your suggestion to explicitly state the methodology used for ncRNA identification and validation, we have considered your feedback carefully. However, we believe that the focus of Section 4 is to present the findings related to ocular diseases, rather than providing detailed methodological information. As such, we have chosen not to add the methodology for ncRNA identification and validation in this section. We have added some technology for ncRNA identification and validation in section 3 and labeled with red.

# The conclusion is clear and summarizes the main findings. It could be enhanced by emphasizing the potential clinical implications of the focus of the review and beneficial in suggesting directions for future research.

Response:

Thank you for your constructive feedback on the conclusion section of our manuscript. We appreciate your insight into the clinical implications and suggesting directions for future research. In response to your suggestion, we have revised the conclusion to place a stronger emphasis on the clinical relevance and directions for future study.

# The review could be more comprehensive by including discussions on the roles of these additional types of ncRNAs in macrophage homeostasis and ocular vascular diseases.

Response:

Thank you for your valuable suggestion regarding the inclusion of additional types of ncRNAs to enhance the comprehensiveness of our manuscript. We understand your point and appreciate your concern for enriching the content of our review. Regrettably, while we recognize the importance of exploring other types of ncRNAs, such as tRNA and rRNA, we found limited research on their involvement in macrophage homeostasis and ocular vascular diseases. This scarcity of studies indicates a promising area for future research and underscores the need for further investigation into the potential roles of these understudied ncRNAs in these contexts.

# Overall, the paper is well-written except few grammatical errors, and the information is presented logically. Additionally, addressing the minor points mentioned above would improve the overall clarity and impact of the paper

Response:

Thank you for highlighting these strengths in our submission. Such recognition encourages us to believe that our work has a foundational value. We appreciate the reviewer' attention to these aspects, which reassures us of the clarity and impact of our initial presentation and visual evidence.

Reviewer 2 Report

Comments and Suggestions for Authors

Dear authors, the work devoted to describing the role of non-coding RNAs in the regulation of macrophage functioning in vascular eye diseases certainly deserves attention. Ophthalmological diseases are a serious problem of modern medicine and the study of the mechanisms of their occurrence and possible ways of influencing the pathological condition is an important task that requires comprehension.

However, there are many questions about the work.

The classification of macrophages used is considered outdated. The text mentions macrophages with other phenotypes many times and does not indicate their correspondence to the types of macrophages described at the beginning of the work.

In describing the functions of different types of macrophages, much attention is paid to their role in the development of tumors, which is understandable, since angiogenesis is an integral part of it. Meanwhile, angiogenesis is a normal process in the body; accordingly, macrophages take part in its regulation and in the absence of pathology. The work does not further mention tumor diseases of the eyes, and therefore constant mention of tumor diseases is unnecessary.

In paragraph 3, the text, figure and caption to the figure repeat each other many times. It is necessary to minimize this paragraph by focusing on the functions of non-coding RNAs specifically in the processes of regulating the functioning and polarization of macrophages.

The above classification of RNA raises questions. The work cited by the authors does not describe these types of RNA. There is no explanation why these three types of RNA were chosen for consideration.

What all the described diseases have in common is pathological angiogenesis. The role of impaired functioning of macrophages in it is undoubted. The mechanisms of this disorder and the influence of non-coding RNAs on this process are very interesting. However, the data presented in the review are not generalized and do not provide a possible answer to this question. Point 4 may need to be reorganized and the data combined based on the involvement of different types of RNA in processes, rather than by disease. This will allow us to identify what is common to all diseases. Also on lines 192-228 there is information about the influence of various factors (but not non-coding RNA) on the functioning of macrophages. These data are shown for patients with macular degeneration. Can they be extrapolated to other diseases? Or should they be removed from the text as not being relevant to the problem being described?

More references should be added to lines 42 and 103. Line 79 should provide a reference to the text that mentions ophthalmological diseases.

On line 101 the link does not match the text.

Line 194-195 says that M2 macrophages induce inflammation, although earlier in the text they are described as anti-inflammatory macrophages.

Author Response

Reviewer 2:

Thank you for your positive feedback and acknowledgment of the importance of our work on elucidating the role of non-coding RNAs in regulating macrophage function in vascular eye diseases. We are grateful for your recognition of the significance of ophthalmological diseases in contemporary medicine and the importance of understanding their underlying mechanisms and potential therapeutic interventions. Your encouraging words inspire us to continue our efforts in contributing to this field of research.

#The classification of macrophages used is considered outdated. The text mentions macrophages with other phenotypes many times and does not indicate their correspondence to the types of macrophages described at the beginning of the work.

Response:

Thank you for your insightful review and for bringing attention to the classification of macrophages used in our manuscript. We introduced the M1 and M2 phenotypes in the section of introduction (line 30-34) and macrophage homeostasis (line71-76). We aimed to provide a comprehensive understanding of macrophage polarization by introducing these classifications along with alternative descriptors such as "proinflammatory type" and "anti-inflammatory type." Our intention in using multiple terms was to enrich the discussion and provide a nuanced portrayal of macrophage phenotypic characteristics. In response to your feedback, we will ensure that the terminology used throughout the manuscript remains consistent and clearly linked to the M1 and M2 classifications introduced in the aforementioned sections. By doing so, we aim to maintain coherence and facilitate better understanding for our readers.

#In describing the functions of different types of macrophages, much attention is paid to their role in the development of tumors, which is understandable, since angiogenesis is an integral part of it. Meanwhile, angiogenesis is a normal process in the body; accordingly, macrophages take part in its regulation and in the absence of pathology. The work does not further mention tumor diseases of the eyes, and therefore constant mention of tumor diseases is unnecessary.

Response:

Thank you for your thoughtful comments regarding the emphasis on tumor-related functions of macrophages in our manuscript. In response to your feedback, we have incorporated a section specifically addressing eye tumor diseases, highlighting the role of macrophages in tumor angiogenesis within ocular tissues.

#In paragraph 3, the text, figure and caption to the figure repeat each other many times. It is necessary to minimize this paragraph by focusing on the functions of non-coding RNAs specifically in the processes of regulating the functioning and polarization of macrophages.

Response:

Thank you for your feedback on paragraph 3 of the manuscript. We have carefully considered your suggestion to minimize repetition by focusing specifically on the functions of non-coding RNAs in regulating the functioning and polarization of macrophages. We believe that emphasizing the functions of non-coding RNAs in the third paragraph could not only highlight the importance of these molecules in cellular processes but also spark readers' interest in new research directions on ncRNA function in macrophage. Additionally, we have mentioned our discussion on the functions of non-coding RNAs in macrophages in lines 135-140.

#The above classification of RNA raises questions. The work cited by the authors does not describe these types of RNA. There is no explanation why these three types of RNA were chosen for consideration.

Response:

Thank you for your attention to citation and your commitment to ensuring the accuracy and clarity of our work. In response to your feedback, we have revisited the citations provided in the manuscript and have made adjustments where necessary to accurately represent the literature. Regarding the selection of the three types of RNA (microRNA, long non-coding RNA, and circular RNA), we chose these based on the relatively extensive body of research available for each type. Given the research available, we believed these RNA types would provide a solid foundation for our review.

#What all the described diseases have in common is pathological angiogenesis. The role of impaired functioning of macrophages in it is undoubted. The mechanisms of this disorder and the influence of non-coding RNAs on this process are very interesting. However, the data presented in the review are not generalized and do not provide a possible answer to this question. Point 4 may need to be reorganized and the data combined based on the involvement of different types of RNA in processes, rather than by disease. This will allow us to identify what is common to all diseases. Also on lines 192-228 there is information about the influence of various factors (but not non-coding RNA) on the functioning of macrophages. These data are shown for patients with macular degeneration. Can they be extrapolated to other diseases? Or should they be removed from the text as not being relevant to the problem being described?

Response:

Thank you for your suggestion about point 4 to reorganize the data based on different pathological processes rather than grouping them by disease. Upon careful consideration, we have decided to maintain the current structure of the fourth section for several reasons. Firstly, the organization by disease allows for a focused and comprehensive examination of the role of non-coding RNAs in specific pathological contexts, which may differ across diseases despite the common feature of pathological angiogenesis. Secondly, restructuring the section based on different pathological processes could potentially lead to fragmentation of the data and make it more challenging for readers to grasp the specific contributions of non-coding RNAs to each disease. We appreciate your valuable input, which has prompted us to carefully reconsider the organization of the manuscript and ensure its clarity and relevance.

Thank you for your thoughtful consideration of the information presented in lines 192-228 regarding the role of macrophages in Age-related Macular Degeneration (AMD).We understand your concern about whether these data can be extrapolated to other diseases or if they should be removed from the text. However, we believe that the discussion of macrophage function in AMD serves as an important foundation for the subsequent exploration of the influence of non-coding RNAs on macrophage function and its implications in AMD pathogenesis. By highlighting the specific context of AMD, we aim to provide a focused and relevant introduction to the role of macrophages in this disease. Subsequently, we transition to the discussion of non-coding RNA regulation of macrophage function and its potential impact on AMD progression. While the data presented may be specific to AMD, the underlying mechanisms of macrophage function and non-coding RNA influence are likely to have broader implications across various diseases characterized by pathological angiogenesis, including but not limited to other ocular vascular diseases.

#More references should be added to lines 42 and 103. Line 79 should provide a reference to the text that mentions ophthalmological diseases.

Response:

Thank you for your valuable feedback on the references in our manuscript. We have added additional references to lines 42 and 103 to provide further support for the statements made in those sections. We have replaced the existing reference with a more relevant citation in line 79.

#On line 101 the link does not match the text.

Response:

Thank you for bringing the discrepancy in line 101 to our attention. We appreciate your careful review of our manuscript. The text at line 101 emphasizes the role of ncRNAs in forming complexes with DNA/RNA and proteins to exert their functions. We deliberately emphasize this point to lay the groundwork for the subsequent discussion on mechanisms.

#Line 194-195 says that M2 macrophages induce inflammation, although earlier in the text they are described as anti-inflammatory macrophages.

Response:

Thank you for bringing this to our attention. After reviewing the original literature, we realized our error. It should have been M1 macrophages inducing inflammation. This has been corrected accordingly.

Round 2

Reviewer 2 Report

Comments and Suggestions for Authors

Dear authors, thank you for the changes made. The additions to the text clear up all the questions I had.

There are only a couple of small comments left.

line 423-429 the text repeats the above.

The eyeball has features in the structure of the immune system. And some properties characteristic of macrophages of other tissues and organs may differ in macrophages of the organ of vision. Are all the properties of macrophages indicated in the text realized in the eyeball?

Author Response

We are honor to receive your positive acknowledgment of our revised manuscript. Thanks for your great questions and our responses are listed as below. We hope our responses could gain your recognition.

line 423-429 the text repeats the above.

Response:

Thanks for your good advice. We have checked our references in line 423-429 and deleted the references which had repeated contents. We have also updated our reference list.

The eyeball has features in the structure of the immune system. And some properties characteristic of macrophages of other tissues and organs may differ in macrophages of the organ of vision. Are all the properties of macrophages indicated in the text realized in the eyeball?

Response:

Thank you for your insightful comment regarding the unique and common features of macrophages in the eye compared to those in other tissues and organs. The eye indeed exhibits unique characteristics within the immune system. Ocular macrophages are divided into tissue-resident macrophages and monocyte-derived macrophages, with both similarities and differences compared to other systems. Specially, ocular macrophages are manifested with immune privilege, and are tend to have a stronger inflammatory response, play an anti-inflammatory role and promote tissue repair, which is crucial to the protection of sensitive tissues like retina. As with other systems of macrophages, ocular macrophages also play the functions of phagocytosis of pathogens, procession and presentation of antigens, and production of inflammatory factors, as well as exhibiting a high degree of diversity and plasticity, in order to alter their functional state and participate in pro-inflammatory, anti-inflammatory, and tissue repair activities. This review focuses primarily on the role of ncRNA in ocular vascular diseases within macrophages. Section 2, "Macrophage Homeostasis and Ocular Vascular Diseases," mainly described the characteristics of changes in macrophage homeostasis in ocular vascular diseases. Section 3, "Functions of NcRNAs," involved the regulatory role of ncRNAs in macrophage homeostasis, which was discussed in a broader perspective, with some studies not yet validated in ophthalmology. The inclusion of these references aims to inspire readers to expand this research within the ophthalmological field. Section 4 details the current research progress on the regulation of ocular vascular diseases by ncRNAs in macrophages, indicating that there is still room for further exploration. Overall, the researches concerned macrophage function in ocular disorders are limited and still need a lot of work to carry out.